# Characterization of a Unique *Bordetella bronchiseptica* vB_BbrP_BB8 Bacteriophage and Its Application as an Antibacterial Agent

**DOI:** 10.3390/ijms21041403

**Published:** 2020-02-19

**Authors:** Mateusz Szymczak, Bartłomiej Grygorcewicz, Joanna Karczewska-Golec, Przemysław Decewicz, Jarosław Adam Pankowski, Hanna Országh-Szturo, Paweł Bącal, Barbara Dołęgowska, Piotr Golec

**Affiliations:** 1Department of Molecular Virology, Institute of Microbiology, Faculty of Biology, University of Warsaw, Miecznikowa 1, 02-096 Warsaw, Poland; mszymczak@biol.uw.edu.pl (M.S.); jaroslaw.adam.pankowski@biol.uw.edu.pl (J.A.P.); h.szturo@student.uw.edu.pl (H.O.-S.); 2Department of Laboratory Medicine, Chair of Microbiology, Immunology and Laboratory Medicine, Pomeranian Medical University in Szczecin, Powstancow Wielkopolskich 72, 70-111 Szczecin, Poland; bartlomiej.grygorcewicz@pum.edu.pl (B.G.); barbara.dolegowska@pum.edu.pl (B.D.); 3Department of Environmental Microbiology and Biotechnology, Institute of Microbiology, Faculty of Biology, University of Warsaw, Miecznikowa 1, 02-096 Warsaw, Poland; karczewska@biol.uw.edu.pl (J.K.-G.); decewicz@biol.uw.edu.pl (P.D.); 4Nalecz Institute of Biocybernetics and Biomedical Engineering, Polish Academy of Sciences, Ksiecia Trojdena 4, 02-109 Warsaw, Poland; bacal@chem.uw.edu.pl; 5Laboratory of Theory and Applications of Electrodes, Faculty of Chemistry, University of Warsaw, 02-093 Warsaw, Poland

**Keywords:** phage therapy, zoonosis, emerging diseases, antibiotic resistance, *Galleria mellonella*, phage stability, biofilm, animal model, atrophic rhinitis, veterinary microbiology

## Abstract

*Bordetella bronchiseptica*, an emerging zoonotic pathogen, infects a broad range of mammalian hosts. *B. bronchiseptica*-associated atrophic rhinitis incurs substantial losses to the pig breeding industry. The true burden of human disease caused by *B. bronchiseptica* is unknown, but it has been postulated that some hypervirulent *B. bronchiseptica* isolates may be responsible for undiagnosed respiratory infections in humans. *B. bronchiseptica* was shown to acquire antibiotic resistance genes from other bacterial genera, especially *Escherichia coli*. Here, we present a new *B. bronchiseptica* lytic bacteriophage—vB_BbrP_BB8—of the *Podoviridae* family, which offers a safe alternative to antibiotic treatment of *B. bronchiseptica* infections. We explored the phage at the level of genome, physiology, morphology, and infection kinetics. Its therapeutic potential was investigated in biofilms and in an *in vivo*
*Galleria mellonella* model, both of which mimic the natural environment of infection. The BB8 is a unique phage with a genome structure resembling that of T7-like phages. Its latent period is 75 ± 5 min and its burst size is 88 ± 10 phages. The BB8 infection causes complete lysis of *B. bronchiseptica* cultures irrespective of the MOI used. The phage efficiently removes bacterial biofilm and prevents the lethality induced by *B. bronchiseptica* in *G. mellonella* honeycomb moth larvae.

## 1. Introduction 

*Bordetella bronchiseptica* is a Gram-negative coccobacilli-shaped bacterium capable of colonizing the respiratory track of mammalian hosts, primarily farm, wild, and companion animals [1]. The bacterium is a common etiological agent of tracheobronchitis, conjunctivitis, rhinitis, mandibular lymphadenopathy, and pneumonia [2]. Its effect tends to be most prominent in tightly packed areas, such as shelters, households, or intensive piggery systems [2]. Interestingly, dogs suffering from *B. bronchiseptica*-associated kennel cough can pass the pathogen to cats [3]. Zoonotic transfer to humans is also possible, albeit rarely reported [4]. The majority of the reported cases included immunocompromised individuals, such as transplant recipients, HIV-infected patients or those with a history of malignancy [5,6,7].

The bacterium spreads via the airborne route to the respiratory ciliated epithelium, often resulting in chronic infections [8]. *Bordetella bronchiseptica* infection may cause a wide range of symptoms, from a mild illness with fever, cough, sneezing, outflow from conjunctiva and swollen lymph nodes, to severe pneumonia with shortness of breath, cyanosis, and death [9]. Infected pigs often suffer from atrophic rhinitis (AR), a costly respiratory disease greatly affecting livestock production. It manifests itself by the atrophy of the nasal turbinate bones and lung lesions. Microscopic lesions are characterized by mucosal neutrophil infiltration, cilia loss, epithelial metaplasia, and bone resorption with replacement by fibrous tissue. Pulmonary lesions include neutrophil infiltration of the airways, necrosis of the alveoli and blood vessels, hemorrhage, and, eventually, extensive fibrosis [10]. In addition to impaired healing, pigs suffering from AR experience decreased appetite and difficulty with food intake, and consequently show a slower increase of body weight [10]. Importantly, *B. bronchiseptica* is known for concurrent infections with *Pasteurella multocida* [11]. In pigs, an infection with *B. bronchiseptica* alone usually leads to mild, nonprogressive atrophic rhinitis (NPAR). A mixed infection with both *B. bronchiseptica* and *P. multocida* results in progressive atrophic rhinitis (PAR) [12]. Whereas dermonecrotic toxin produced by *P. multocida* is the main factor behind the observed damage in pigs, the presence of *B. bronchiseptica* promotes the rate of nasal cavity colonization by the toxigenic bacterium [13].

Currently, veterinary vaccines are a standard method of preventing *B. bronchiseptica*-associated diseases in animals. While multiple vaccination methods have been developed, the existence of sufficient duration of immunity is being questioned [14]. Additionally, due to the appearance and dissemination of antibiotic resistant *B. bronchiseptica* strains [15], alternative means of combating *B. bronchiseptica* infections are required. One report shows a case of woman who suffered from recurring *B. bronchiseptica* infections, due to bacterial persistence. The relapses occurred despite minocycline treatment [16]. A suggested explanation for the observed persistence was the *B. bronchiseptica* ability to create biofilm. The formation of biofilm has also been proposed to justify the decreased efficiency of vaccines used against *B. pertussis*—a pathogen closely related to *B. bronchiseptica* [1]. This further underlines the necessity of developing new methods of fighting *Bordetella*-associated infectious diseases, especially the strategies that could efficiently reduce a threat from biofilm production. With their natural ability to prey on specific strains of bacteria, bacteriophages (phages) have been postulated an efficient solution. Phages active against human pathogenic bacteria have been studied at least since the beginning of the 20th century [17]. Even though *B. bronchiseptica* species has been known for more than a hundred years, there have only been few reports describing its bacteriophages [18,19,20,21,22,23].

In this study, we isolated, described the biology of, as well as explored the therapeutic potential of a new *B. bronchiseptica* bacteriophage vB_BbrP_BB8, further referred to as BB8, which had been isolated from a sewage sludge sample. Beyond the standard characterization and testing, the new phage was evaluated *in vivo* in a *Galleria mellonella* model for its ability to kill *B. bronchiseptica* cells. Not only did the phage prove capable of reducing the viability of bacterial biofilm, it also almost completely eliminated the lethality of infection in the *G. mellonella* infectious disease model.

## 2. Results

### 2.1. Isolation of the BB8 Phage

The *B. bronchiseptica* A5 phage host strain—isolated in this study from a chronically diseased companion animal—was identified based on MALDI-TOF mass spectrometry profiling (confidence score of 2.447) and 16S rRNA gene sequencing (GenBank acc. no. MT040725, 100% identity to 16S rRNA gene from *B. bronchiseptica* strain LSBSG1, acc. no. MN082542). The strain displayed natural resistance to cephalosporins (cephalexin, cefuroxime) and amoxicillin/clavulanic acid.

The phage was isolated from an environmental sample with a standard enrichment technique [24] using the above mentioned *B. bronchiseptica* host strain. The phage was named vB_BbrP_BB8, according to the guideline of Adriaenssens and Brister [25]. In this work, the phage will be further referred to as BB8.

Apart from the BB8 ability to infect *B. bronchiseptica* we did not observe its efficiency in producing plaques on any other tested organism. During the host range analysis, we investigated more than 20 animal isolates of *Klebsiella* spp., *Acinetobacter* spp., *Enterobacter cloacae*, and *Escherichia coli*.

The BB8 phage formed clear plaques of approximately 1 mm in diameter. Over time the plaque size grew to 3 mm (Figure 1C–E). The presence of a halo zone around plaques suggested release of lytic enzymes during infection.

Transmission electron microscopic images revealed a icosahedral virion of approximately 37 nm in diameter, typical of the *Podoviridae* family in the order *Caudovirales* (Figure 1A,B).

### 2.2. Phage BB8 Resistance to Physico-Chemical Factors

To evaluate the phage potential to be used in therapy and other applications, phage activity and stability under the various conditions important from the viewpoint of practical use were determined. The BB8 phage displayed high stability at pH 4 to 8, but lost it quickly at lower pH values. In addition, whereas there was no immediate loss of activity after increasing the pH above 8, a substantial activity decrease in an alkaline environment was observed over time (Figure 2A). Furthermore, BB8 was sensitive to increased temperature, experiencing a serious decrease in activity over time at 43 °C. Higher temperatures (60–80 °C) resulted in an immediate loss of activity (Figure 2B). It is noteworthy that the phage lysate did not display any losses in stability when stored at 4 °C for at least 6 months. Furthermore, BB8 proved susceptible to UV radiation (Figure 2C). Within the first minute of exposure, almost half of the phage particles lost activity. Within 20 min, the number of active phage particles dropped to a nearly undetectable level.

### 2.3. Adsorption, One-Step Growth and Collapse Assays

The phage adsorption to the host cell is the first step towards successful infection and propagation. An adsorption kinetics analysis indicated that approximately 96% of BB8 particles adsorbed to the surface of *B. bronchiseptica* cells within the first minute. The value increased to over 99% after 5 min (Figure 3A). Efficient adsorption enables a phage to infect a bacterial cell and replicate by hijacking its metabolism. To further explore BB8 development in *B. bronchiseptica*, a one-step growth assay was employed (Figure 3B). The observed eclipse time was 55 ± 5 min and the latency period was 75 ± 5 min. The yield of phages per infected cell (the burst size) was approximately 88 ± 10 phage particles. Additionally, the BB8 phage proved highly efficient in destroying liquid bacterial cultures, irrespective of the MOI used. A complete lysis was observed with all MOI variants tested (Figure 3C). While high MOIs could cause the collapse after just 2 h, a much lower MOI showed effects after 4–5 h (Figure 3C).

### 2.4. Phage Genome Exploration

Therapeutic applications of a phage candidate require an in-depth exploration of the phage genome. The BB8 phage contains dsDNA genome (41,593 bp). The fully annotated sequence of the BB8 phage was deposited in the GenBank database (acc. no. MK984681). The correctness of assembly of the genome sequence was verified through digestion of genomic DNA, isolated from the phage virions, with a series of restriction enzymes and comparison of the resulting pattern with that obtained by *in silico* digestion of the assembled BB8 genome sequence. The pattern of restriction fragments on an agarose gel image was in line with that in the *in silico* generated image, thus confirming the determined BB8 sequence (Figure 4).

The genome analysis revealed that BB8 genome possesses a structure similar to that of the T7 lytic phage (Figure 5A). However, the BB8 nucleotide (nt) sequence shared no significant similarities to the sequence of T7, and the BB8 amino acid (aa) sequence showed identity of less than 55% in a few regions (Figure 5B). Likewise, no similarities to BB8 at either nt or aa sequence levels were found in the other *Podoviridae* phages capable of infecting *B. bronchiseptica* (i.e., BMP-1, BIP-1, and BPP-1) (Figure 5B). Interestingly, no gene that could potentially encode RNAP was identified in the nt sequence of any of the three phages. These three phages share a high (>86%) level of sequence identity between each other, but display no such relation to T7-like viruses. Furthermore, BB8 itself shows an aa sequence similarity (of up to 85% in some regions) with two *Ralstonia* phages (DU-RP-I and RsoP1EGY). Particularly striking is the nt and aa sequences identity of their RNAPs (Figure 5B).

Importantly, we did not identify any genes coding for toxins, components of lysogenic development pathway (with the exception of an integrase gene, a putative relic, discussed below) or any other genes that would disqualify BB8 from therapeutic applications.

### 2.5. Applicability of BB8 Phage as an Antibacterial Agent

To evaluate the potential of the BB8 phage to act as a treatment for *B. bronchiseptica* infections we analyzed its abilities to destroy bacterial biofilm *in vitro* and to fight *B. bronchiseptica* infection in an *in vivo* honeycomb moth (*G. mellonella*) larvae model. The results show that the phage effectively destroys *B. bronchiseptica* biofilm (Figure 6). An established, 48-h biofilm was treated with BB8 for 4 h, after which there was a statistically significant (*p* < 0.05) reduction in the biofilm viability and biomass (Figure 6A,B). The high efficiency in biofilm viability reduction was observed with all phage concentrations tested (Figure 6A). The viability reduction level reached approx. 92% when phages in 1 × 10^7^ and 1 × 10^5^ PFU/mL concentrations were applied. In the case of 1 × 10^3^ PFU/mL phage concentration, viability reduction of around 86% was observed. Additionally, the treatment resulted in approx. 75%, 71%, and 59% of biofilm biomass reduction for phage concentrations of 1 × 10^7^, 1 × 10^5^, and 1 × 10^3^ PFU/mL, respectively (Figure 6B). To confirm these effects, the bacterial biofilm exposed to BB8 was further analyzed by SEM (Figure 6C–D). A lower number of bacterial cells (compared with a control variant of the biofilm without phage treatment) together with completely lysed cells were visible on the SEM images of *B. bronchiseptica* biofilm treated with BB8. Moreover, a changed morphology was observed for a number of bacterial cells. The cells were not as round as those in the control biofilm, which suggests that they were dead.

Experiments on the larvae model infected with *B. bronchiseptica* showed that BB8 phage treatment can drastically reduce the mortality of the infected organism (Figure 7). First, the most effective lethal dose of bacteria was identified (Figure 7A). Injection with 2.5 × 10^7^ CFU of *B. bronchiseptica* per larva (Figure 7A) proved most lethal and was further used in BB8 phage treatment of *B. bronchispetica*-infected larvae. Approximately 70% of untreated larvae died within 24 h after having been injected with the pathogen cells and the entire population was gone after 96 h. Conversely, the phage-treated, *B. bronchiseptica*-infected larvae displayed only a minimal drop in survivability over the entire experiment duration (survivability rate was ≥ 90% after 96 h). Additionally, the injection of buffer alone or the phage suspension alone (control groups) caused no trauma to larvae. Importantly, the presence of the phage in haemocoel caused neither stress symptoms (including melanization) nor larvae mortality.

## 3. Discussion

Antibiotic-resistant bacterial infections are a major threat to not only human but also animal health. As recognized by the One Health concept, human and animal health are in fact interconnected through—among other things—the biology and ecology of their shared pathogens. *B. bronchiseptica*, an emerging zoonotic pathogen, is a common cause of infections among diverse domestic, farm, and wild animals, but is most important and best described in dogs and pigs [2]. In humans the actual burden of diseases caused by *B. bronchiseptica* is unknown, but it has been postulated that *B. bronchiseptica* infections are more common than previously appreciated, and that some hypervirulent isolates may in fact be responsible for undiagnosed respiratory infections [26]. In humans, the commonly used diagnostic methods that employ PCR-based identification accurately detect *B. pertussis* and *B. parapertussis*, but not *B. bronchiseptica* [26].

While *B. bronchiseptica* is generally susceptible to several groups of antibiotics, including tetracyclines, β-lactams, macrolides, aminoglycosides, and sulphonamides, the effects tend to differ between isolates [26,27]. Aminoglycoside and aminocyclitol, phenicol, fluoroquinolone, β-lactam, sulfonamide, tetracycline, and trimethoprim resistance mechanisms were described in *B. bronchiseptica* isolates [15]. Most of the identified antibiotic resistance genes in *B. bronchiseptica* were located on conjugative plasmids. The rising level of antibiotic resistance in *B. bronchiseptica* strains combined with the ease of acquiring and transferring resistance genes from and to, other bacterial genera that share the same niches, especially *E. coli* [15], highlight the need for developing effective treatment strategies against this pathogen other than those based solely on antibiotics.

One promising alternative to antibiotic treatment is phage therapy [17], in which the activity and specificity of bacterial viruses is harnessed to destroy pathogenic bacteria [28]. In this work, we isolated a new phage—BB8—infecting a *B. bronchiseptica* strain that had been isolated from an animal infection. The phage has a lytic lifestyle, efficiently adsorbs to its host cells and produces a high yield of daughter particles (approx. 90 PFU/IC). Due to these traits, we explored the potential of BB8 as a weapon against *B. bronchiseptica* infections in detail. A substantial reduction of bacterial biofilm viability and biomass were obtained upon exposing it to the phage lysate (approx. 70% of biofilm biomass removal and approx. 90% biofilm viability reduction). While the ability of phages to reduce the biomass of bacterial biofilms was previously demonstrated in multiple bacterial models [29], this is the first report of *B. bronchiseptica* biofilm removal by a bacteriophage. Phage-mediated biofilm removal may be critically important in therapeutic applications as—according to the National Institutes of Health (NIH)—biofilms are implicated in over 80% of microbial infections in the body (NIH, 2002). Therapies that aim at biofilm removal are vital in the case of *B. bronchiseptica* and other pathogens for which biofilm formation is a major virulence factor.

Laboratory conditions are different from those observed in real animal infection sites. Therefore, therapeutic potential of a phage candidate should be evaluated in *in vivo* models that can mimic the natural environment of an infection. In this study, BB8 phage displayed the ability to almost completely eliminate the lethality of *B. bronchiseptica*-infected honeycomb moth larvae—an increasingly popular infectious disease model. The larvae immune system is structurally and functionally similar to the innate immune response of mammals. The major advantages to using the insect model are also reduced laboratory animal use, lowered cost, and the ease of manipulation in high containment. Importantly, the results obtained using the insect show strong correlation with those obtained using mice or mammalian cells [30,31]. The honeycomb moth larvae infection model was successfully applied in the past for pathogenesis studies, assessment of activity and toxicity of new antimicrobial agents, and for the evaluation of phage therapy. Specifically, the model was used for the assessment of *in vivo* therapeutic effectiveness of phages against *Cronobacter sakazakii*, *Pseudomonas aeruginosa*, *Burkholderia cepacia*, and *Acinetobacter baumannii* infections [32,33,34,35]. Recently the larvae model has been used along a mouse model of acute pneumonia for the evaluation of phage treatment of carbapenem-resistant *Acinetobacter baumannii* (CRAB) infections. A single dose of phage Bϕ-R2096 at the very high MOI of 100 increased survival rate of CRAB-infected honeycomb moth larvae from 0% to 50% (24 h post CRAB infection). Survival rates of phage-treated, CRAB-infected mice were 30% (at MOI of 0.1) and 100% (at MOI of 10) at day 10 post CRAB infection. Mice in the bacteria-only-treatment group all died by day 5 after infection with CRAB [35].

Melanization of the larvae hemolymph is a marker of the humoral immune responses in *G. mellonella* and is analogous to abscess formation in mammalian infections [36]. The lack of melanization after injection of BB8 phage in this study proved that BB8 did not react with the immune system of *G. mellonella* larvae. This, together with the other, above-mentioned attributes, makes BB8 a safe therapeutic candidate for combating *B. bronchiseptica* infections.

Genome analysis of BB8 provided further insight into the nature of this phage. Genes responsible for virus-host interactions, cell lysis, and virus assembly were identified. Genes encoding RNA and DNA polymerases were also found in the BB8 genomic sequence. The presence of an endolysin gene in the BB8 genome explains the formation of a halo zone around BB8 plaques, as the encoded enzyme degrades bacterial cell wall [37]. Another interesting feature is the presence of a potential integrase gene. Identification of the gene was based on the presence of multiple integrase or tyrosine site-specific recombinase domains (e.g., cd00796 or COG4974 with e-values of 2.31e-33 and 1.48e-09, respectively) as well as sequence homology to Enterobacteria phage lambda integrase (1Z1B_A—100% HHpred probability). The BB8 seems to be a strictly lytic phage, as it produced only clear plaques and completely lysed the host culture irrespective of MOI used for infection, culture OD (from 0.2 to 0.6), or temperature (30 and 37 °C). Therefore, the presence of a potential integrase gene could be explained as an evolutionary relic or an element acquired through horizontal gene transfer. Additionally, no other gene encoding DNA-binding protein—a potential repressor—was identified.

Despite having the overall genome structure typical of T7-like phages, BB8 displays no significant similarity to T7 at the level of genomic sequence. However, it shows similarity (of less than 40%) of its RNAP amino acid sequence to that of the enzyme found in T7-like viruses. Other *Podoviridae* phages known to infect *B. bronchiseptica* form a separate group, even less similar to T7 at the level of their nt and aa sequences (possible similarity of less than 20%), and are characterized by the lack of genes encoding RNAP. Overall, BB8 shares the highest amino acid sequence similarity (85% in the most similar regions) with *Ralstonia* phages. The *Ralstonia* spp. are mainly known as plant pathogens [38], but they can also act as opportunistic human pathogens [39]. Importantly, neither the *in silico* analysis of the high-quality whole genome sequence nor the phage development assays showed in BB8 a functional lysogenic lifestyle, virulence factors, toxins, resistance genes, or transducible elements, thus indicating the safety of the phage for therapeutic applications.

## 4. Materials and Methods

### 4.1. Sample Origin

The strain of *B. bronchiseptica* was isolated on March 06, 2014 from a nasal swab obtained from a 4-year-old, male Belgian hare (*Oryctolagus cuniculus*) with a chronic rhinorrhea (runny nose), in the course of routine treatment. The BB8 phage was isolated from a wastewater sample collected from a sewage treatment plant in Wołomin (Wołomin County, Masovian Voivodship, Poland) on August 25, 2018 according to [24].

### 4.2. Bacterial Host and Culture Conditions

The identity of the bacterial strain as *B. bronchiseptica* was determined with the use of MALDI-TOF profiling (MALDI Biotyper system, Bruker Daltonik GmbH, Bremen, Germany) and sequencing of the 16S rRNA gene (primer sequences: 27F 5′-AGAGTTTGATCMTGGCTCAG-3′ and 1492R 5′-GGTTACCTTGTTACGACTT-3′) by commercial services. LB, LB-agar, top-agar (LB medium with the addition of 0.7 % of agar) and Brain Heart Infusion (BHI) media (Merck KGaA, Darmstadt, Germany) were used for cultivation of *B. bronchiseptica* at 37 °C.

### 4.3. Phage Isolation

Phage isolation was performed according to [24]. After the isolation, three rounds of propagation from an individual phage plaque were carried out to prepare a lysate of the pure phage strain.

### 4.4. Phage Propagation and Purification

Phage BB8 was propagated in a *B. bronchiseptica* liquid culture. The culture was infected during the exponential growth phase (OD_600_ = 0.2 − 0.5) with one phage plaque or with the phage lysate to reach the multiplicity of infection (MOI) = 0.01. After the phage addition, the culture was incubated with shaking at 37 °C until complete lysis of bacteria was observed. Phage lysate was then centrifuged (8000 *g*, 10 min, room temperature). The number of plaque forming units (PFUs) was determined by a standard plaque technique, as described previously [40]. Briefly, double-layer LB agar plates with an overnight culture of *B. bronchiseptica* in top agar were used and 2.5 µL of phage lysate serial dilutions was spotted onto a bacterial lawn. Plates were incubated at 37 °C and plaques were counted after 16 h.

### 4.5. Thermal and pH Stability of the Phage

For pH stability testing, 10 µL of the phage lysate (1 × 10^10^ PFU/mL) was mixed with 990 µL LB broth in a series of tubes, each with a different pH (adjusted using NaOH or HCl), and incubated for 10, 30, or 60 min at room temperature (RT). For temperature stability testing, 10 µL of the phage lysate (1 × 10^10^ PFU/mL) was mixed with 990 µL of LB broth in a series of tubes, and the samples were incubated at 43, 60, or 80 °C for 15, 30, or 60 min. At each of these time points, the number of phages (plaque forming units, PFUs) was calculated. Control phage samples were incubated at pH 7 and RT, respectively.

### 4.6. Phage Resistance to UV Light

For UV resistance testing, 10 µL of the phage lysate (1 × 10^10^ PFU/mL) was mixed with 990 µL of LB broth. The mixture was then spotted onto a polystyrene Petri dish and exposed to UV-C light (UVC lamp, Philips TUV-30-W-245 nm Lamp, type No. 57413-P/40, Philips co., Warszawa, Poland) from a distance of 65 cm for 1, 2.5, 5, 10, 15, 25, and 30 min. A control sample was incubated on a laboratory bench at RT.

### 4.7. Transmission Electron Microscopy (TEM)

Transmission electron microscopic images of phage virions were obtained using TEM LIBRA 120 (Carl Zeiss NTS GmbH, Oberkochen, Germany) microscope (HT = 120 kV, LaB6 cathode). Samples were prepared by applying a droplet of phage suspension to thin, carbon-coated copper grids (400 mesh), followed by immersing the grids in 1% uranyl acetate for contrasting. Then, the samples were left to dry at RT.

### 4.8. Adsorption Kinetics

The adsorption assay was conducted as described previously [40]. Briefly, 1 mL of an overnight *B. bronchiseptica* culture was spiked with 10 µL of the phage suspension to reach MOI = 0.1. The mixture was incubated at 37 °C. After 1, 2.5, 5, 10, and 15 min, 100-µL aliquots were withdrawn and centrifuged (5000 *g*, 1 min, RT) to deposit the phage-adsorbed cells as sediment. The titer of the remaining free phages was determined by supernatant titration on double-layer agar plates. The initial number of phages (100% of the phages used) was determined by adding an appropriate volume of the BB8 phage lysate to a medium without bacteria, followed by titration. The number of adsorbed phages was determined as a decrease in PFUs in the supernatant relative to the initial number of phages.

### 4.9. One-Step Growth Curve

For one-step growth analysis, 25 mL of LB medium was inoculated with an overnight culture of *B. bronchiseptica* (50:1, *v*/*v*). The culture was grown until OD_600_ = 0.2 and then infected with the phage at MOI = 0.1 and incubated at 37 °C for 1 min (theoretically 96% of phages should have adsorbed). Then, 25 µL of infected cells was transferred to 25 mL of fresh LB broth and incubated at 37 °C with shaking (150 rpm). At appropriate timepoints, 300 µL and 300 µL of samples treated with 300 µL of chloroform were individually collected for PFUs calculations. The samples for estimation of the number of infection centers (ICs) were analyzed as described previously [41].

### 4.10. Bacterial Culture Collapse

For the host culture collapse studies, an overnight culture of bacteria was refreshed in LB medium and allowed to grow until reaching OD_600_ of 0.2. Then each, culture was infected with the phage at MOIs ranging from 0.0001 to 1. The OD_600_ measurement was made every 15 min till the lysis of bacterial culture was observed. Measurements were carried out with the use of Sunrise absorbance microplate reader and Magellan v 6.6 2009 software (Tecan Trading AG, Männedorf, Switzerland).

### 4.11. Biofilm Formation and Assessment of Phage Lytic Activity in Biofilm

The *B. bronchiseptica* biofilm formation protocol was adapted from [42] with minor modifications. Briefly, overnight cultures of *B. bronchiseptica* were diluted 1:100 into fresh BHI medium and 200 µL of a culture per well was transferred to 96-well polystyrene microtiter plates and incubated without agitation at 37 °C for 48 h with medium change every 6 h. Post incubation, wells were washed with PBS buffer to remove planktonic and unattached cells. Established biofilm was treated with phage suspensions (final concentration of approx. 1 × 10^7^, 1 × 10^5^, and 1 × 10^3^ PFU/mL) in BHI medium and incubated for 4 h at 37 °C. After phage treatment, cell viability and biomass in biofilm were measured using resazurin and crystal violet staining assays according to [43] with minor modifications. The PBS-washed wells containing *B. bronchiseptica* biofilm were filled with 200 µL of a fresh BHI medium containing 20 mM of resazurin and incubated for 40 min at 37 °C with shaking (100 rpm). After incubation, the medium was transferred for fluorescence (λex = 520 nm; λem = 590 nm) measurement. Biofilm was washed using PBS and fixed with ethanol for biomass quantification. Then ethanol was removed and plates were air dried. Staining was conducted by the addition of crystal violet (1% *w*/*v*) and incubation for 20 min at RT. Stained biofilm was washed and 200 µL of ethanol:acetone (8:2, *v*/*v*) solution was added to each well for decolorization. The biofilm was then quantified by determining the optical density of the resulting solution at 595 nm. All of the spectrometric measurements were carried out with the use of EnVision 2105 multimode plate reader (PerkinElmer, Waltham, MA, USA).

### 4.12. Scanning Electron Microscopy (SEM)

Visualization of biofilms was carried out by scanning electron microscopy (SEM), as described before [44]. Biofilms were fixed for 24 h in 0.1 M cacodylate buffer (pH 7.3) with 3% glutaraldehyde followed by washing for 60 min with cacodylate buffer without glutaraldehyde, and then four times for 30 min with the fresh buffer, followed by dehydration for 6 h in 96% ethanol. Biofilms were air-dried and coated with gold-palladium (2–4 nm thick) and analyzed at nanometer image resolution by MERLIN SEM (Carl Zeiss NTS GmbH, Oberkochen, Germany) at accelerating voltages in the range of 2–5 kV.

### 4.13. In Vivo Assessment of Phage Activity in B. bronchiseptica-Infected Galleria mellonella Larvae

The *in vivo* test on *G. mellonella* larvae was performed as described previously [45] with minor modifications. Briefly, to assess *B. bronchiseptica* pathogenicity, larvae were infected with three different dose of bacteria (2.5 × 10^3^, 2.5 × 10^5^ and 2.5 × 10^7^ CFU/larva), by an injection of 5 µL dose into the larval hemolymph behind the last proleg and incubated at 37 °C for 120 h. To evaluate the bacteriophage activity, 2.5 × 10^7^ CFU/larva bacterial dose was chosen. Larvae were infected with *B. bronchiseptica* by an injection 5 µL dose into the larval hemolymph behind the last proleg. After 20 min, the treated larvae received the bacteriophage suspension at MOI = 1 or MOI = 10, which was injected (5 µL) into the opposite side to the bacterial infection site. Injections with a buffer control and phage suspension control were also carried out in control larvae groups to assess the negative impact of an injection and the phage toxicity. Larvae were incubated at 37 °C for 96 h. The larvae of a dark color that did not respond to physical contact were marked as deceased.

### 4.14. Phage DNA Extraction and Restriction Enzyme Digestion

Phage DNA was isolated with the use of Genomic Mini AX Phage kit (A&A Biotechnology, Gdynia, Poland). For restriction analysis, the phage DNA was digested with the following single FastDigest enzymes: *Nco*I and *Nde*I or with the following enzyme combinations: *Nco*I + *Sma*I and *Nco*I + *Nde*I. Enzymes and buffers were obtained from Thermo Fisher Scientific (Waltham, MA, USA). Restriction digest was performed in accordance with the manufacturer’s protocols. Restriction fragments were analyzed on 1% agarose gel and stained with ethidium bromide.

### 4.15. Whole Genome Sequencing (WGS)

The complete nucleotide sequence of the BB8 phage was determined in the DNA Sequencing and Oligonucleotide Synthesis Laboratory (oligo.pl) at the Institute of Biochemistry and Biophysics, Polish Academy of Sciences (Warsaw, Poland). The phage genome was sequenced on an Illumina MiSeq platform in a paired-end mode with a v3 chemistry kit. The sequence reads obtained were filtered for quality and assembled using Newbler v3.0 software (Roche, Basel, Switzerland).

### 4.16. Bioinformatics

The annotation of BB8 genome was carried out manually with Artemis software [46]. Gene calling was performed using PHANOTATE with tRNAScan-SE [47,48]. The functional annotation of encoded proteins was based on homology searches performed using BLAST programs [49] against NCBI protein non-redundant or virus databases. Moreover, the identification of protein domains was performed with CD-Search and HHpred tools [50,51].

The comparative genomic analysis of BB8 was performed with Circoletto tool [52] and the construction of a sequence similarity network. The following thresholds were applied with the Circoletto tool: e-value of 1e-30 when comparing nucleotide sequences of phage genomes, and e-value of 1e-10 and query coverage of HSP of at least 75% when comparing predicted proteomes of analyzed phages. The construction of a similarity network was based on all-against-all BLASTp comparison of BB8 with all 10,362 bacterial viruses available at NCBI (as of February 01, 2019). During the BLASTp search, e-value of 1e-10, query coverage of HSP of at least 75%, and at least 50% of sequence identity thresholds were applied. The results were then parsed using Python script and visualized as a network on which each node represents a phage genome and each edge, and its thickness, correspond to the number of proteins shared between two phages above the set thresholds. The networks were visualized in Gephi 0.9.2 using ForceAtlas 2 layout [53,54].

The *in silico* electrophoretic gel image was prepared with the use of SnapGene Viewer 3.2.1 software (GSL Biotech, Chicago, IL, USA; available at snapgene.com).

### 4.17. Statistical Analysis

The biofilm assay data were analyzed with one-way ANOVA. Results were considered statistically significant at a *p* value of <0.05. The Kaplan–Meier survival curves for the *G. mellonella* model were prepared. All statistical analyses were carried out using GraphPad Prism 5.02 (Graph Pad Software, San Diego, CA, USA).

## 5. Conclusions

In conclusion, a new phage infecting *B. bronchiseptica* was isolated and characterized. We demonstrated its effective development in host cells not only under the standard laboratory conditions, but also in a bacterial biofilm and in an *in vivo* animal infection model. Importantly, the *in vivo* study revealed no mortality or side effects in the phage-treated honeycomb moth larvae group. The genome analysis of BB8 revealed that it is a unique phage with no sequence similarity to other known *Bordetella* phages. Taken together, the results of an *in vitro*, *in vivo* and *in silico* integrated approach indicated that phage BB8 is a safe and effective treatment alternative to the standard, antibiotic-based therapies of *B. bronchiseptica* infections, which—based on current knowledge—are probably more common than previously thought, with some hypervirulent isolates responsible for undiagnosed respiratory infections also in humans [26].

## Figures and Tables

**Figure 1 ijms-21-01403-f001:**
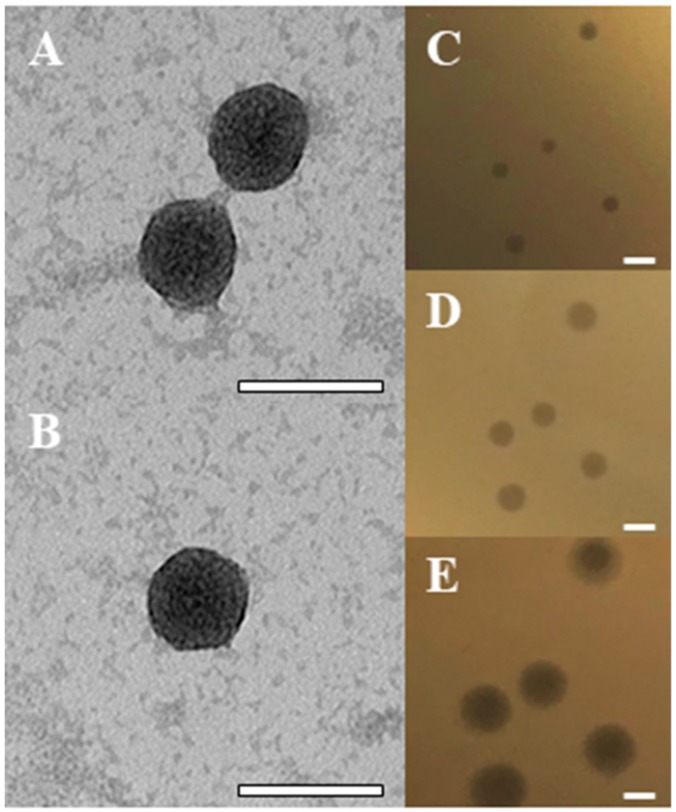
Characteristics of BB8 phage virions and plaques. (**A** and **B**) TEM image of BB8 phage virions is typical of the *Podoviridae* family, *Caudovirales* order. The scale bars in **A** and **B** represent 50 nm. (**C**–**E**) The growth of phage plaques over time. Plaque phenotype observed after 8, 12, and 24 h is presented in panels **C**–**E**, respectively. The scale bars in **C**–**E** panels represent 3 mm.

**Figure 2 ijms-21-01403-f002:**
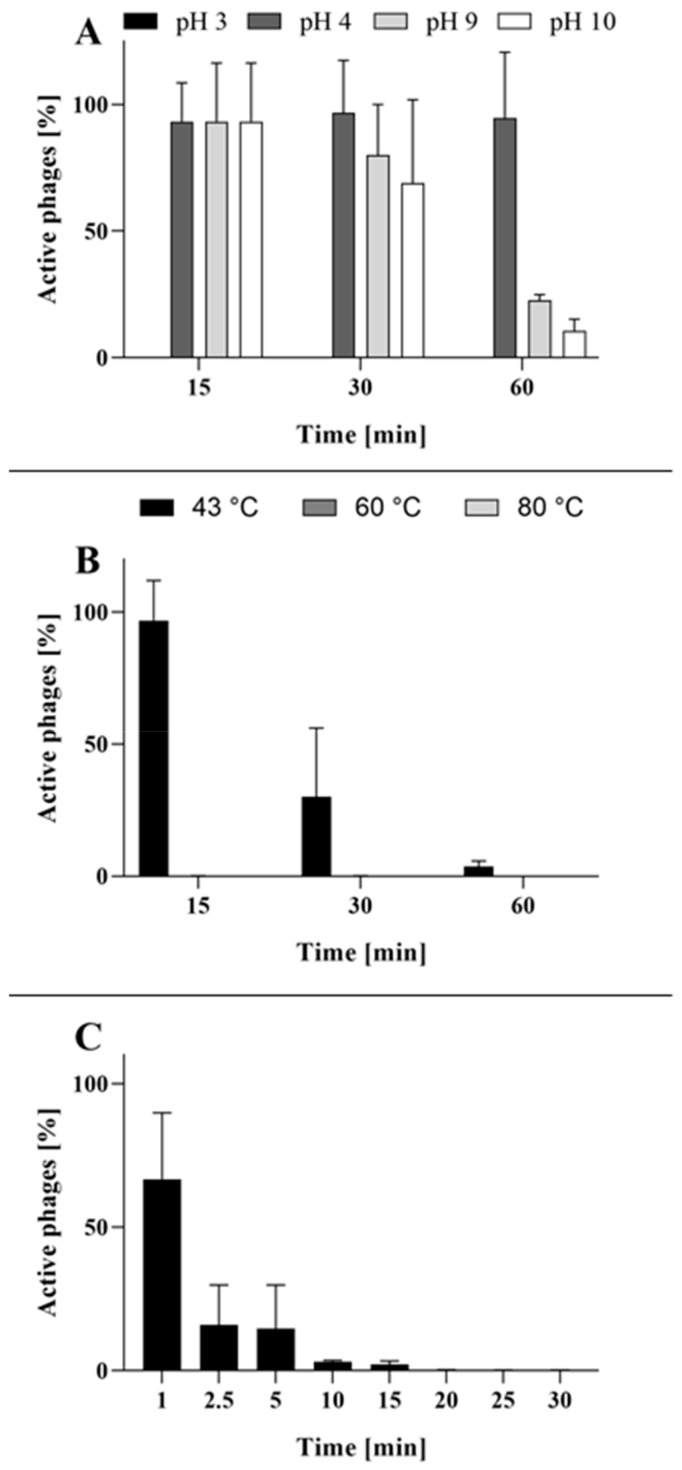
Phage BB8 stability under various conditions. (**A**) Effect of pH on the phage activity. The number of active phage particles at pH 3 was below the detection limit (<400 PFU/mL). (**B**) Effect of temperature on the phage activity. The number of active phage particles at 60 °C and 80 °C was below the detection limit (<400 PFU/mL). (**C**) Effect of UV radiation on the phage activity. The presented results are average values from three experiments, with SD represented by error bars.

**Figure 3 ijms-21-01403-f003:**
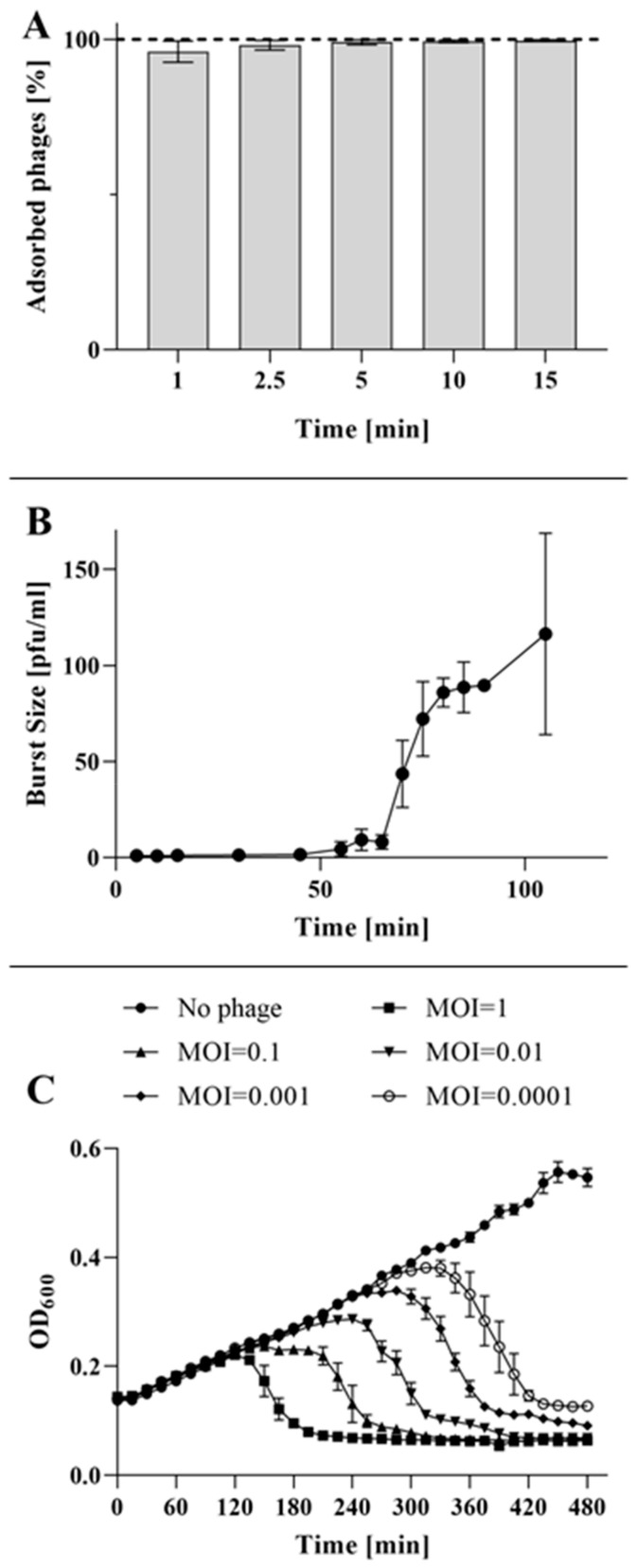
Phage BB8 development. (**A**) Adsorption of phage particles to the bacterial cells was assessed after 1, 2.5, 5, 10, and 15 min. (**B**) One-step growth curve shows that the eclipse period was a little under 60 min while latent period was around 75 min and resulted in the production of approx. 88 phage particles. (**C**) Collapse assay of the phage-infected bacterial culture indicated the high efficiency of host cell lysis. Regardless of the MOI used, the bacterial culture was completely lysed by the phage. The presented results are average values from three experiments, with SD represented by error bars.

**Figure 4 ijms-21-01403-f004:**
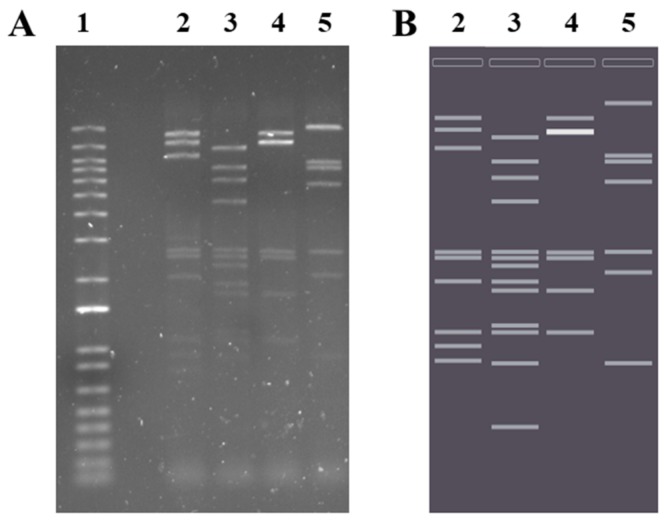
Analysis of restriction patterns of BB8 genome. (**A**) Electrophoretic gel image of BB8 genomic DNA digested with various restriction enzymes shows the same pattern as (**B**) *in silico* prepared electrophoretic gel image of the restriction fragments. Lines: (1) DNA size marker, (2) *Nco*I + *Sma*I, (3) *Nco*I + *Nde*I, (4) *Nco*I, and (5) *Nde*I.

**Figure 5 ijms-21-01403-f005:**
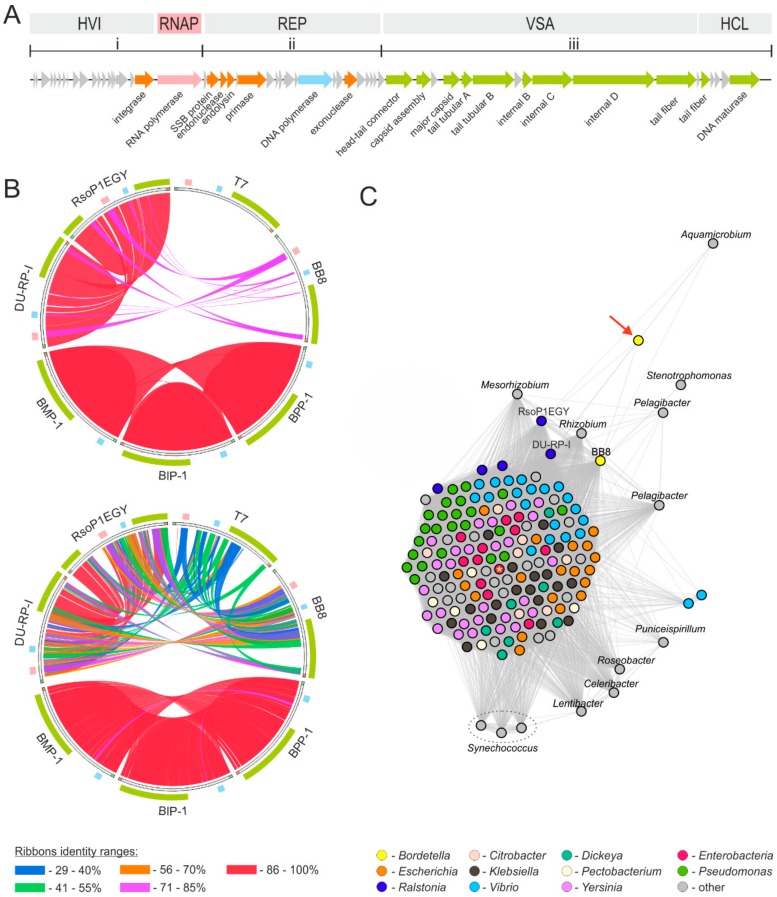
Genome organization and comparative genomics of BB8 phage. (**A**) Genome organization of BB8. A block at the top shows host-virus interaction (HVI), RNA polymerase (RNAP), DNA replication (REP), virus structure and assembly (VSA) and host cell lysis (HCL) modules, as well as transcription of genes of class i, ii, and iii. The modules were identified and described with the use, and according to, the genome organization of the model phage T7. The arrows below represent open reading frames identified in the BB8 genome and their transcriptional orientation. The coloring scheme of the arrows is as follows: grey—hypothetical proteins with unknown function, orange—proteins with predicted function, pink—RNA polymerase, blue—DNA polymerase, green—virus structure and assembly proteins. (**B**) The comparison of BB8 sequences with those of T7 phage of Enterobacteria (NC_001604), two most similar *Ralstonia* phages, i.e., RsoP1EGY (MG711516) and DU-RP-I (MF979559), and all known podoviruses infecting *Bordetella*, i.e., BMP-1 (AY526908), BIP-1 (AY526909), and BPP-1 (AY029185). The comparison was performed with the Circoletto tool on nucleotide (top graph) and amino acid (bottom graph) sequence levels. The blocks on the outmost ring correspond to the location of RNAP, DNA polymerase, and VSA modules. The colors of these correspond to the colors in panel A. (**C**) Sequence similarity network of proteins encoded by BB8 and the most closely related bacterial viruses. Each node represents a single phage genome and each edge, and its thickness, reflects the number of similar proteins encoded by two phages above the set thresholds. The coloring is based on the host’s family taxonomy. Enterobacteria T7 phage’s node is additionally indicated with a yellow star. All viruses within the presented network belong to *Podoviridae* family except the *Bordetella* phage vB_BbrM_PHB04 (MF663786) annotated as a member of *Siphoviridae* family—indicated by the red arrow.

**Figure 6 ijms-21-01403-f006:**
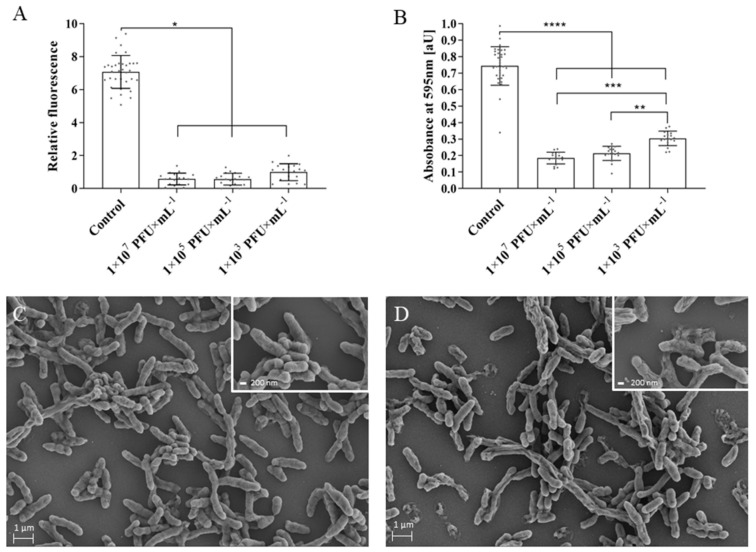
Effect of phage BB8 treatment on *B. bronchisepica* biofilm. Decrease of (**A**) viability and (**B**) mass of *B. bronchiseptica* biofilm after 4-h treatment with BB8 phage in three different concentrations. A *B. bronchiseptica* biofilm without the addition of BB8 served as a control. Each dot in graphs (**A**) and (**B**) represents a biological replicate and the results were analyzed with one-way ANOVA (**** *p* < 0.0001; *** *p* < 0.001; ** *p* < 0.01; * *p* < 0.05). The SEM images of *B. bronchiseptica* biofilm untreated and treated with phage BB8 are presented in panels **C** and **D**, respectively. Figure insets present untreated (**C**) and phage-treated (**D**) biofilms at higher magnification. The scale bars in **C** and **D** represent 200 nm.

**Figure 7 ijms-21-01403-f007:**
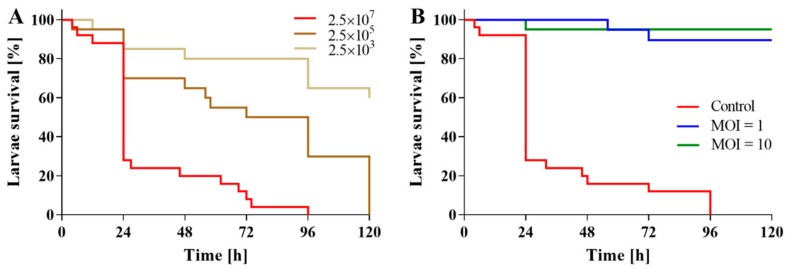
Lethality of honeycomb moth larvae. (**A**) Influence of *B. bronchiseptica* cell number on larvae survivability. Twenty-five larvae per group and three groups (2.5 × 10^3^, 2.5 × 10^5^, and 2.5 × 10^7^ CFU/larva) were analyzed. (**B**) Treatment of *B. bronchiseptica*-infected *G. mellonella* larvae with BB8 phage. Control group (red line) includes *Galleria* larvae after treatment with *B. bronchiseptica* alone (approx. 2.5 × 10^7^ CFU). Treatment with phage BB8 almost completely abolished the lethality of the infection at MOIs of 1 and 10 (blue and green line, respectively). Twenty-five larvae per group and three groups (control, MOI = 1, and MOI = 10) were analyzed. Survivability in larvae treated with buffer alone or with phage alone was 100% (these lines were omitted for clarity).

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
