# Peer review of "Characterization of a Unique Bordetella bronchiseptica vB_BbrP_BB8 Bacteriophage and Its Application as an Antibacterial Agent"

_ijms, 2020, doi:10.3390/ijms21041403_

Round 1

Reviewer 1 Report

This is a description of a new phage infecting Bordetella bronchiseptica, a significant pathogen in swine and potentially in humans, with antibiotic resistance. A phage therapy approach to treatment would be valuable, and therefore this paper has a high level of significance and interest. The paper is well written and organized clearly. 

The presence of a putative integrase gene is surprising. As the authors indicate, this may be a relic or randomly acquired. However, it would be important to confirm the lytic life cycle. There is no description in the methods of how the genome annotation was performed, including gene function prediction. What is the evidence that this is an integrase. Is there another gene encoding a DNA-binding protein that is a potential repressor?  The authors say in line 307 "no lysogenic cycle was observed for the BB8 phage under the tested conditions". Are they referring to the standard plaque assays - looking for turbid plaques - or did they specifically test for lyoseny? The latter would require more focused experiments. Temperate phages may not produce lysogens in sufficient quantity to be observed as turbid plaques. I would not expect this type of phage to be temperate, but since that is a significant characteristic for application in phage therapy it seems important to clarify.

I do not find the SEM images in Fig. 6 compelling for demonstration of cell viability. The visual difference in the cells is subtle. A fluorescence microscopy method such as PI/SYTO9 might be better to visualize cell death in the biofilm.

For Fig. 7, the text says that controls with buffer alone and phage suspension alone were performed. It seems like both controls should be included on graph B. I assume it would be 100% survival across that time frame, but it should be included for completeness. Although they did not find toxin, etc. genes, the function of all genes cannot be predicted, so any data documenting safety should be included.

Are the sequences publicly available? I could not find accession numbers for the phage genome or bacteria 16s sequence.

Author Response

Point 1: The presence of a putative integrase gene is surprising. As the authors indicate, this may be a relic or randomly acquired. However, it would be important to confirm the lytic life cycle. There is no description in the methods of how the genome annotation was performed, including gene function prediction. What is the evidence that this is an integrase. Is there another gene encoding a DNA-binding protein that is a potential repressor? The authors say in line 307 "no lysogenic cycle was observed for the BB8 phage under the tested conditions". Are they referring to the standard plaque assays - looking for turbid plaques - or did they specifically test for lyoseny? The latter would require more focused experiments. Temperate phages may not produce lysogens in sufficient quantity to be observed as turbid plaques. I would not expect this type of phage to be temperate, but since that is a significant characteristic for application in phage therapy it seems important to clarify.

Response 1: We thank the Reviewer 1 for pointing out these imprecise descriptions in the manuscript. We have added the following information:

(lines: 457-461) Materials and Methods section: a description of how the annotation was performed, (lines: 311-320) how the integrase gene was identified, and that no another gene encoding a DNA-binding protein (a potential repressor) was found. Additionally, we rewrote the sentence about the lytic character of the BB8 phage

We believe that these parts of the manuscript are now clear.

Point 2: I do not find the SEM images in Fig. 6 compelling for demonstration of cell viability. The visual difference in the cells is subtle. A fluorescence microscopy method such as PI/SYTO9 might be better to visualize cell death in the biofilm.

Response 2: We thank the Reviewer 1 for pointing out this issue. We added insets to the Figure 6 that present bacterial biofilm at higher magnification. We believe that in this form the differences are more visible.

Point 3: For Fig. 7, the text says that controls with buffer alone and phage suspension alone were performed. It seems like both controls should be included on graph B. I assume it would be 100% survival across that time frame, but it should be included for completeness. Although they did not find toxin, etc. genes, the function of all genes cannot be predicted, so any data documenting safety should be included.

Response 3: We thank the Reviewer for the suggestion. Yes, survival of the larvae after treatment with buffer alone or phage suspension alone was 100%. Figure 7 looked confusing when control lines were included. However, we added appropriate data to Figure 7 caption.

Point 4: Are the sequences publicly available? I could not find accession numbers for the phage genome or bacteria 16s sequence.

Response 4: Many thanks for pointing out this omission. We have added accession numbers for both phage BB8 genome (lines: 157-158) and B. bronchiseptica 16s rRNA gene (line 95).

Reviewer 2 Report

The manuscript represents the work devoted to exanibarion of  Bordetella Bronchiseptica vB_BbrP_BB8 Bacteriophage and its antibacterial properties. 

The study was performed at a good modern level and comprehensively characterizes the bacteriophage . The article is written clearly and well illustrated. 

I have no comments on the work and the article. 

Author Response

Point 1: The study was performed at a good modern level and comprehensively characterizes the bacteriophage . The article is written clearly and well illustrated. I have no comments on the work and the article.

Response 1: We thank the Reviewer for appreciating our work. We did our best to provide a clear description and presentation of the results in the manuscript.